# Influence of Industrial Wastewater Irrigation on Heavy Metal Content in Coriander (*Coriandrum sativum* L.): Ecological and Health Risk Assessment

**DOI:** 10.3390/plants12203652

**Published:** 2023-10-23

**Authors:** Ilker Ugulu, Zafar Iqbal Khan, Abdulwahed Fahad Alrefaei, Shehnaz Bibi, Kafeel Ahmad, Hafsa Memona, Shahzadi Mahpara, Naunain Mehmood, Mikhlid Hammad Almutairi, Aima Iram Batool, Asma Ashfaq, Ijaz Rasool Noorka

**Affiliations:** 1Faculty of Education, Usak University, Usak 64000, Turkey; 2Department of Botany, University of Sargodha, Sargodha 40100, Pakistanasmaashfaq@uos.edu.pk (A.A.); 3Department of Zoology, College of Science, King Saud University, Riyadh 11451, Saudi Arabiamalmutari@ksu.edu.sa (M.H.A.); 4Department of Zoology, Queen Mary College, Lahore 54000, Pakistan; 5Department of Plant Breeding and Genetics, Ghazi University, Dera Ghazi Khan 32200, Pakistan; 6Department of Zoology, University of Sargodha, Sargodha 40100, Pakistan; 7Department of Veterinary Medicine, University of Sassari, 07100 Sassari, Italy; 8Department of Plant Breeding and Genetics, College of Agriculture, University of Sargodha, Sargodha 40100, Pakistan; ijaznoorka@uos.edu.pk

**Keywords:** health risk, pollution, trace metal, vegetable

## Abstract

The primary objective of this study was to determine the heavy metal contents in the water–soil–coriander samples in an industrial wastewater irrigated area and to assess the health risks of these metals to consumers. Sampling was done from areas adjoining the Chistian sugar mill district Sargodha and two separate sites irrigated with groundwater (Site 1), and sugar mill effluents (Site 2) were checked for possible metal contamination. The water–soil–coriander continuum was tested for the presence of cadmium (Cd), cobalt (Co), chromium (Cr), copper (Cu), iron (Fe), manganese (Ni), lead (Pb), and zinc (Zn). The mean concentrations of all metals were higher than the permissible limits for all studied metals except for Mn in the sugar mill wastewater, with Fe (8.861 mg/L) and Zn (9.761 mg/L) exhibiting the highest values. The mean levels of Fe (4.023 mg/kg), Cd (2.101 mg/kg), Cr (2.135 mg/kg), Cu (2.180 mg/kg), and Ni (1.523 mg/kg) were high in the soil at Site 2 in comparison to the groundwater irrigated site where Fe (3.232 mg/kg) and Cd (1.845 mg/kg) manifested high elemental levels. For coriander specimens, only Cd had a higher mean level in both the groundwater (1.245 mg/kg) and the sugar mill wastewater (1.245 mg/kg) irrigated sites. An estimation of the pollution indices yielded a high risk from Cd (health risk index (HRI): 173.2), Zn (HRI: 7.012), Mn (HRI: 6.276), Fe (HRI: 1.709), Cu (HRI: 1.282), and Ni (HRI: 1.009), as all values are above 1.0 indicating a hazard to human health from consuming coriander irrigated with wastewater. Regular monitoring of vegetables irrigated with wastewater is strongly advised to reduce health hazards to people.

## 1. Introduction

The significance of food safety is increasingly growing in people’s lives, especially in their dietary choices. While packaged foods are generally seen as more unsettling in terms of safety in daily nutrition routines, it should not be forgotten that it is important for vegetables and fruits to go through safe and healthy processes from seed to table [1]. Metal contamination in agricultural soils is an important issue regarding food safety and potential hazards to human health [2,3]. There are many sources of trace elements and heavy metal accumulation in vegetables, but soil particularly stands out as an important heavy metal collector [4,5]. These trace elements and heavy metals accumulate in the vegetables and enter the human body when the vegetables are consumed [6].

Geological and anthropogenic activities, such as small-scale industry, military operations, the use of agricultural pesticides, and industrial wastes, are the main sources of heavy metal pollution [7,8]. Despite the fact that the sugar industry is seasonal and only works 150–160 days a year, the process of making sugar generates a substantial amount of waste, including organic materials and pulp suspended in the pressing sludge and several xenobiotics including heavy metals [9,10]. For this reason, sugar industry facilities, like other industrial facilities, cause trace element and heavy metal pollution in their impact areas if required precautions are not taken [11]. In regions such as Pakistan where irrigation is done with wastewater due to the scarcity of freshwater resources and the impacts of climate change, the use of industrial wastewater in agricultural irrigation causes trace element and heavy metal pollution in crops [12]. Therefore, it is possible that vegetables and other agricultural products grown in these regions may pose a threat to human health due to heavy metal accumulation [13]. The widespread dispersion of heavy metal contamination has disrupted the ecosystems and presents significant health risks to human populations [12,13].

Coriander (*Coriandrum sativum* L.) belongs to the Apiaceae family. In Asia, this herb is colloquially called Chinese parsley, fresh coriander and dhania, whereas it is also known as cilantro in America. All parts of the plant are edible, but the dried seeds and fresh leaves are most commonly used in cooking [14]. For example, chopped coriander is used for garnishing in Chinese, Vietnamese, and Thai dishes. In South Asian dishes, fresh coriander leaves are used to make dipping sauces and salads. Coriander is naturally rich in vitamins (vitamin C, vitamin A, and vitamin K) and other dietary minerals like iron, magnesium, calcium, and manganese [15].

Since wastewater irrigation is a widespread practice in Pakistan, research on the impacts of this sort of irrigation on the environment and heavy metal contamination is frequently conducted using this method [16,17,18]. The translocation of heavy metals through the food chain is one of the consequences of soils polluted with heavy metals and may result in health issues in humans [9]. The increasingly bioavailable fraction of heavy metals necessitates risk assessments in agricultural areas. Although the risks of heavy metal accumulation in various agricultural products and the use of wastewater irrigation in Pakistan have been the subject of many studies, no comprehensive research has been found on coriander. The goals of this study were to determine the heavy metal contents in the water–soil–coriander samples in industrial wastewater irrigated areas and to assess the health risks of these metals to people. Even low amounts of heavy metals can be deposited in the crops which requires a continuous monitoring of the movement of these metals through the food chain.

## 2. Materials and Methods

### 2.1. Study Area

The district of Sargodha is bounded by the district of Jhelum to the north, the Chenab River to the east, and the district of Jhang to the south (Figure 1). The Sargodha district includes five sub-districts, namely, Kot Momin, Sillanwali, Sahiwal, and Shahpur, and has a total size of 5854 square kilometers. The average recorded temperatures in the Sargodha district are 35–45 °C in the summer and 12 °C in the winter.

### 2.2. Collection of Samples

The area around Chishtian Sugar Mill Limited was chosen as the sampling area for the study. The sugar mill is located in the Farooka village of the Sillanwali sub-district. It was founded in 1990 and has an installed cane-crushing capacity of 7000 TCD (tons of cane per day). 

Two crop-cultivated sites around the sugar mill utilizing groundwater (Site 1; tube-well irrigation), located at 31°49′29.7″ N 72°25′40.4″ E, and wastewater from the sugar industry (Site 2), located at 31°52′13.5″ N 72°27′43.7″ E, as irrigation sources, were tested, and their metal concentrations were determined. Water was taken from both sources in polypropylene bottles thoroughly washed with distilled water and 1 mL of HNO_3_ that was added to stop microbial development. The bottles were placed in a freezer before further examination.

Twenty-five (25) soil samples each were obtained from sites 1 and 2 that were respectively irrigated by groundwater and wastewater from the sugar industrial effluent. Soil samples were randomly collected from the uppermost 25 cm of the soil’s surface. After being dried and crumpled, all materials were crushed through 2 mm crush strainers. Until analysis, soil samples were kept on croft paper.

The whole coriander plant was selected for the analysis. The vegetable specimens (n = 50) were collected from the same sites where soil samples were obtained. The coriander samples were cleansed in deionized water to remove any impurities before being dried at 80 °C to a uniform weight.

### 2.3. Sample Preparation

For the digestion of water samples, 5 mL of trace metal grade nitric acid (HNO_3_), obtained from Merck (Darmstadt, Germany), was added to the sample contained in the digestion tube at 80 °C for 45–55 min. A clear solution obtained after digestion was filtered and transferred to the bottles for further analysis.

Initially, 10 mL of trace metal grade HNO_3_, obtained from Merck (Germany), was added to one gram of soil in a beaker for digestion. The mixture left overnight was stirred the next day and put in the digestive tube for an hour at 150 °C. An amount of 5 mL of trace metal grade hydrogen peroxide (H_2_O_2_; Merck, Germany) was then added to the mixture. The *v*/*v* ratio of HNO_3_ and H_2_O_2_ was 2:1. The material was removed from the digestion tube once the process of digestion was complete and a clear solution was obtained. The mixture was filtered using filter paper and then mixed with distilled water to create a volume of 50 mm.

One gram of each coriander specimen was placed in a digestive tube together with 10 mL of nitric acid (HNO_3_) and kept overnight. The digestion tube was placed on a heated plate the next day at 150 °C. After 35 min, 5 mL hydrogen peroxide (H_2_O_2_) was added to the tube, and the resulting solution was heated until it was clear. The digested material was removed from the digestion tube once digestion was complete and filtered using Whatman No. 42 filter paper, and then given a 50 mm volume increase with deionized water.

### 2.4. Analysis of Physicochemical Properties of Soil Samples

The three physical and chemical characteristics of soil that were examined were electrical conductivity (EC), pH, and organic matter (OM). The pH of the soil was determined using a pH meter [19]. Based on Richard [20], an electrical conductivity estimate was performed. The OM of the soil was measured using the Walkley and Black acid digestion procedure [21].

### 2.5. Metal Analysis

The samples of water, soil, and vegetables were further processed for analysis of Cd, Co, Cr, Cu, Fe, Ni, Pb, and Zn using the Perkin-Elmer AAS-300 atomic absorption spectrophotometer (AAS). Limit of Detection (LOD) values were evaluated based on acknowledged practices described in the literature [22]. The value was classified as LOD since the blank solution’s standard deviation (SD) and signal-to-noise ratio were both determined to be 3. The operating conditions of the flame atomic absorption spectrometry for the relevant heavy metals are shown in Table 1. The presence of nickel (Ni) and chromium (Cr) was determined using the gaseous hydride generation technique.

### 2.6. Quality Control

The calibration of the instrument used diagnostic marker standardization values from Merck (Germany). Throughout the investigation, deionized water was meticulously used to clear the crystalline pupillages. A statement of value was completed, and the findings were checked for consistency using Specialized Position Quantifiable assessments (SRM-2711 for soil and SRM NIST 1570a for trace elements in spinach leaves). The mean SRM recoveries for Pb, Cu, Co, Mn, Cd, Cr, Zn, and Fe in soil were 107%, 101%, 96%, 106%, 93%, 95%, and 97%, respectively. The mean SRM recoveries for these metals in coriander were 98%, 104%, 96%, 99%, 91%, 95%, and 90%, respectively.

### 2.7. Statistical Analysis

The significant differences in the metal levels between the two irrigation sites was estimated using one-way ANOVA with SPSS 24. The statistical significance of the discrepancies in the results was examined at the 0.05, 0.01, and 0.001 levels of significance [23]. Hierarchical Clustering Analysis in IBM SPSS 24 software was also used to analyze and contrast the associations between metal values in the samples.

### 2.8. Bioconcentration Factor (BCF)

The BCF values were calculated using the following formula:BCF = C_veg_/C_soil_

The abbreviation C_veg_ (mg/kg, dry weight) stands for the concentration of metals in plant tissues, whereas C_soil_ (mg/kg, dry weight) is used to describe the concentration of metals in soil [24].

### 2.9. Enrichment Factor

The concentrations of metals deposited in plants are compared to the levels in the soil using the enrichment factor (EF). For this assessment, the following formula is used:EF = C_plant_ × C_ref.plant_/C_soil_ × C_ref.soil_

The metal concentrations in the plant and soil samples used in the study are represented in this formula by C_plant_ and C_soil_, respectively, while the standard metal concentrations in the plant and soil are represented by C_ref.plant_ and C_ref.soil_ [25].

### 2.10. Daily Intake of Metals

One technique for identifying consumer-related health risks from food consumption is daily metal intake (DIM). The DIM values in this study were determined in accordance with Sajjad’s [26] definition:DIM = C_metal_ × D_food intake_/B_average weight_

### 2.11. Health Risk Index

Health risk index (HRI) identifies whether consuming contaminated food poses a risk to a person’s health. It was used in this study to estimate the potential metal exposure that would result from consuming the coriander samples [25,27]:HRI = DIM/RfD
where RfD is the oral reference dose of metal as specified by USEPA. Each metal has its own defined dose.

## 3. Results and Discussion

### 3.1. Heavy Metal Contents in Irrigation Water Samples

The metal values in the irrigation water samples varied from 0.719 to 1.537, 0.108 to 0.315, 0.534 to 0.86, 0.527 to 0.991, 3.223 to 8.861, 0.702 to 0.963, 1.323 to 1.968, 0.031 to 0.138, and 2.528 to 9.761 mg/L for Cd, Co, Cr, Cu, Fe, Mn, Ni, Pb, and Zn, respectively (Figure 2). All metals, except for Fe and Zn, did not have statistically significant differences between the values as per the statistical analysis (*p* > 0.05, 0.01, and 0.001) (Table 2).

Many countries, especially those with limited access to clean water, irrigate field crops, horticulture, and vegetables with water that is of poor quality. Smallholder farmers also use reclaimed water at various purification levels to irrigate crops and vegetables, and for forage in many arid and semi-arid lands [28,29]. The heavy metal values in this study were higher than the maximum limits in water established by FAO, WHO, and Standard Guidelines in Europe [30], except for Mn. Although the elevated metal concentrations in wastewater from the sugar industry are of industrial origin, they might also be caused by other factors such as road traffic, urban runoff, and aerosol particles [31].

### 3.2. Biochemical Composition of Soil and Its Heavy Metal Content

The metal concentrations in the soil samples ranged from 1.845 to 2.101, 0.735 to 0.785, 1.166 to 2.135, 1.492 to 2.180, 3.232 to 4.023, 0.665 to 0.868, 1.340 to 1.523, 0.327 to 0.382, and 0.420 to 0.806 mg/kg for Cd, Co, Cr, Cu, Fe, Mn, Ni, Pb, and Zn, respectively (Figure 3). Except for Co, Pb, and the values in the groundwater and the wastewater from the sugar industry, there was no significant effect of site on any of the metal values in the soil samples (*p* > 0.05) (Table 3).

Heavy metals are by nature non-biodegradable. Because they linger in the soil for a long time, they eventually move up the food chain to the plants and endanger the health of both animals and people [32]. The metal values in this study were lower than the maximum limits for Ni (9.06), Cr (9.07), Pb (3.50), Fe (56.9), Co (9.1), Mn (46.74), Cu (8.39), and Zn (44.19) mg/kg [33] while values of Cd (1.96 to 2.01 mg/kg) were higher than the maximum limit (1.49 mg/kg) [34]. Numerous studies that were conducted using various plant specimens, locations, wastewater types, seasons, and environments under local conditions produced results for Cd that were similar to those of this study but also above the allowable limit supporting the findings of this study [18,24,25].

Both sample areas’ soils were found to be loamy and had a pH of 7.83 (Site 1) and 7.45 (Site 2). The soil samples’ electrical conductivities were found to be 1.78 dsm^−1^ and 5.09 dsm^−1^ in Site 1 and Site 2 respectively. Additionally, the percentage of organic matter in the soil samples was 0.55% for Site 1 and 0.69% for Site 2, respectively (Table 4).

The solubility and bioavailability of heavy metals in the soil can be influenced by physicochemical characteristics like pH, redox potential, cation exchange capacity, and organic matter content [35]. Long-term irrigation of soil with untreated wastewater alters its physicochemical characteristics and raises the concentration of heavy metals in the soil [36]. Heavy metals are more mobile due to the high acidity of the soil, and they are also more bioavailable due to the reduction of the redox potential in the soil, which turns insoluble heavy metal ions into soluble forms [37]. The soil surface is one of the most important variables influencing the availability of metals in the soil, according to Khan et al. [12]. It was determined that Site 2, which received wastewater irrigation, had a slightly lower pH value. This might be because wastewater irrigation results in the decomposition of organic materials and the formation of organic acids in the soil [32]. Furthermore, Siddique et al. [38] found that locations that were watered with wastewater had relatively larger quantities of organic matter than other sites. Electrical conductivity has an impact on the development of salinity, which is the most important indication in wastewater irrigation fields [39]. Crops accumulate more Cd from soils having low cadmium content, indicating its bioavailability and translocation ability to the plants [40].

### 3.3. Heavy Metal Contents in Coriander Samples

The heavy metal values in the groundwater and sugar industry wastewater irrigated coriander samples ranged from 1.245 to 1.623, 0.025 to 0.220, 0.460 to 0.548, 0.775 to 1.403, 0.926 to 1.595, 0.395 to 1.026, 0.697 to 0.865, 0.077 to 0.216, and 1.642 to 2.103 mg/kg for Cd, Co, Cr, Cu, Fe, Mn, Ni, Pb, and Zn, respectively (Figure 4). The statistical analysis revealed that with the exception of Ni, Pb, and Cr, the differences in the metal accumulation values between the coriander samples watered with groundwater and wastewater were statistically insignificant (*p* > 0.05, 0.01, and 0.001) (Table 5).

The high metal concentration in the medium may, in general, lead to an increased transfer to plants. In many countries, wastewater irrigation is the primary method of delivering heavy metals and other contaminants to crops [40]. The current concentrations for Co, Cr, Cu, Fe, Ni, Pb, Zn, and Mn in the coriander samples were lower than the maximum allowable limits of Co (50 mg/kg), Cr (2.3 mg/kg), Cu (73.3 mg/kg), Fe (425.5 mg/kg), Mn (500 mg/kg), Ni (67 mg/kg), Pb (0.3 mg/kg), and Zn (99.4 mg/kg) as reported by FAO/WHO [39]. Nevertheless, the Cd content (1.25 to 1.41 mg/kg) in coriander samples was above the maximum permitted limit (0.2 mg/kg) as indicated by FAO/WHO [39]. Air dust from industries, phosphate fertilizers, and irrigation wastewater are the primary sources of Cd [41]. The high Cd readings in coriander samples can be associated with the listed sources. Rai et al. [40] concluded that soil qualities, physicochemical features of irrigated wastewater, and metal properties all play a role in the complicated process of metal and other pollution uptake by plants.

### 3.4. Assessment of Heavy Metal Pollution and Associated Health Risks

#### 3.4.1. Hierarchical Clustering Analysis

The studied metals in the soil were divided into two major groups on the basis of average linkage cluster analysis, as shown in the dendrogram. Eight metals clustered in the second main group indicating similar accumulation behavior in soil, while only Fe clustered separately in the first main group (Figure 5). Average linkage cluster analysis of metal values in the coriander samples also showed two main groups in the dendrogram (Figure 6). The first main group included Fe, Cu, Zn, and Cd, whereas the second group had the remaining five metals. In the first main group, Zn was separated from the other three metals. The other metals are in the second of the two subgroups that were formed from the second major group composed of Pb and Co (Figure 6).

Unlike soil samples, the inclusion of zinc in plants may be influenced by the traits of the plant species as well as Zn’s mobility from the soil to the plants. In their study on the effects of different organic fertilizers on the accumulation of heavy metals in vegetables in Sargodha, Pakistan, Ugulu et al. [6], they reached the conclusion that hierarchical cluster analysis could discriminate the accumulation of Fe and Zn from that of other metals. In this context, the outcomes and breadth of the present study are comparable to those of Ugulu et al. [6]. In studies on heavy metals in vegetables in India, Bhatia et al. [42] found that Zn and Cd had a higher transfer factor than other metals because of their enhanced mobility from soil to the edible parts of plants.

#### 3.4.2. Bioconcentration Factor

Except for the Cr and Zn concentrations, the BCF values for the metals at Site 2 irrigated with the wastewater from the sugar industry were higher than those at Site 1 irrigated with groundwater. The BCF for Co was the lowest, whereas the BCF for Zn was the greatest. The orders of the BCF values for the heavy metals at Site 1 and Site 2 were Zn > Mn > Ni > Cu > Cr > Zn > Fe > Pb > Co and Zn > Mn > Cd > Cu > Ni > Pb > Fe > Cr > Co, respectively (Table 6). When describing the bioavailability of metals at a specific place in a plant species, the bioconcentration factor can also be utilized [43]. In the present investigation, Zn was determined to have the highest BCF. The high Zn content seen in the plants can be explained by the relative abundance of the metals in the deeper soil layers and the earth’s crust [44]. On the other hand, a long-term increase in the Zn concentration and various heavy metals might occur when the wastewater is used for irrigation in agriculture [45]. The considerable BCF values for Zn reported in this experiment lend credence to these conclusions.

#### 3.4.3. Enrichment Factor

The EF values at both sites were in the descending order of Cd > Ni > Zn > Mn > Cr > Pb > Cu > Fe > Co. The highest EF value was found for Cd at Site 2 where wastewater from the sugar industry was used, and the lowest EF value was found for Co at Site 1 (Table 7). The enrichment factor is linked to a number of variables, such as edaphic factors, the amount of metals in the environment, the plants’ capacities to absorb metals, physiological makeup, and growth phenomena [46,47]. Cd had the highest EF value in the current research (25.75) recorded at Site 2. The EF values for Mn, Ni, and Zn also proceeded over 1.0. In Bhakkar, Pakistan, Khan et al. [16] studied the accumulation of metals in Luffa samples (*Luffa cylindrica* (L.) Roem.) and discovered EF values over 1.00 for Pb, Zn, and Cd. The fact that metal transport and accumulation, from soil to root, root to stem, and in grains, vary from site to site in plants, it can be the cause of the variations in the results [48,49].

#### 3.4.4. Daily Intake of Metals and Health Risk Index

Zn had the highest DIM value, which was recorded at Site 2, at 2.103 mg/kg/day, while Pb had the lowest value, which was determined at Site 1. In terms of HRI values, Cd at Site 2 had the greatest value (173.2), whereas Co at Site 1 had the lowest value (0.001) (Table 8).

Human exposure to potentially harmful metals, such as As, Cd, and Pb, can result in metabolic problems and cancer risks even at low concentrations [50]. The maximum daily metal intake (DIM) value in this study was calculated as 2.103 for Zn in the Site 2 region based on Pakistan Bureau of Statistics (PBS) data, which assumes that an average person weighs 55.9 kg and consumes 0.345 kg of vegetables per day [11]. In spinach specimens cultivated in Beijing, China, Khan et al. [51] discovered DIM values of 0.032, 0.008, 0.002, 0.0003, 0.005, and 0.005 for Zn, Cu, Pb, Cd, Cr, and Ni, respectively. In comparison to the values obtained by Khan et al. [51], the current values were greater. The variances in the results can be due to the properties of the wastewater as well as the regional features and plant types. Wastewater irrigation results in increased daily elemental intake, particularly in leafy vegetables [52,53,54,55].

Using more wastewater for irrigation is a very important factor that promotes the accumulation of metals in the soil and boosts their bioavailability to plants. This situation increases the health risks associated with metal absorption into vegetative components [56,57,58]. Health risk index (HRI) values for the coriander samples from the areas irrigated with groundwater and wastewater from the sugar sector varied from 0.001 to 173.2. Co, Cr, and Pb had HRI values below 1.0 and did not seem to be a hazard to human health; in contrast, Cd, Ni, Fe, Mn, Cu, and Zn constituted a substantial health risk. The Cd buildup with HRI values > 1 (173.2) suggested that this metal is probably going to have a very bad effect on local health. The multi-tissue carcinogen cadmium (Cd) is highly toxic to both humans and animals [59,60,61,62,63].

## 4. Conclusions

The goals of this study were to determine the heavy metal contents in the water–soil–coriander samples in industrial wastewater irrigated areas and assess the health risks of these metals to people. The heavy metal readings in the irrigation water used in the region, with the exception of Mn, exceeded the authorized maximum levels. However, it was discovered that the amounts of heavy metals in the soil and coriander samples irrigated with these waters were lower than the maximum permitted limits, with the exception of Cd. This suggests that utilizing water from the sugar industry, which has a specific quantity of heavy metals, poses a risk to consumers. In contrast to Cd, Ni, Fe, Mn, Cu, and Zn, which seemed to be a health risk, Co, Cr, and Pb had HRI values below 1.0. Cd buildup with HRI values > 1 (173.2) suggested that this metal is probably going to have a very bad effect on local health. Around the world, particularly in developing nations, wastewater irrigation is widely used. Therefore, it is advised that wastewater treatment plants be developed and properly utilized to minimize danger. The adoption of appropriate bioremediation techniques and the cultivation of plants with reduced accumulations may be advantageous in areas where these opportunities are scarce. In any case, future research should focus on the possible health risks associated with exposure to heavy metals through various pathways. The study does not identify the probable sources of metal contamination, apart from industrial wastewater, which is a limitation of this study.

## Figures and Tables

**Figure 1 plants-12-03652-f001:**
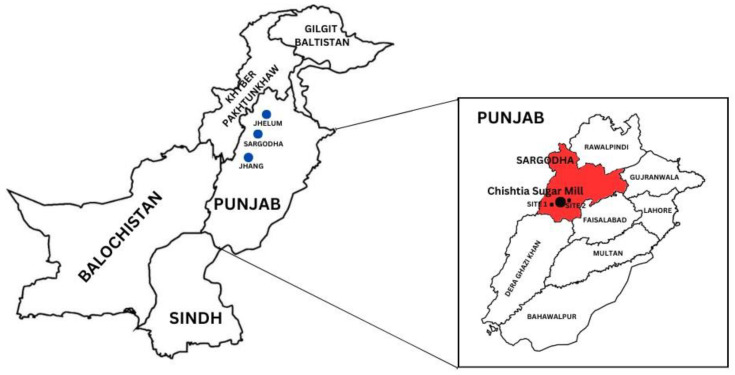
Map of Pakistan showing the location of Sargodha and its neighboring districts. Inset shows Chistian Sugar Mill in district Sargodha, and other divisions of Punjab province.

**Figure 2 plants-12-03652-f002:**
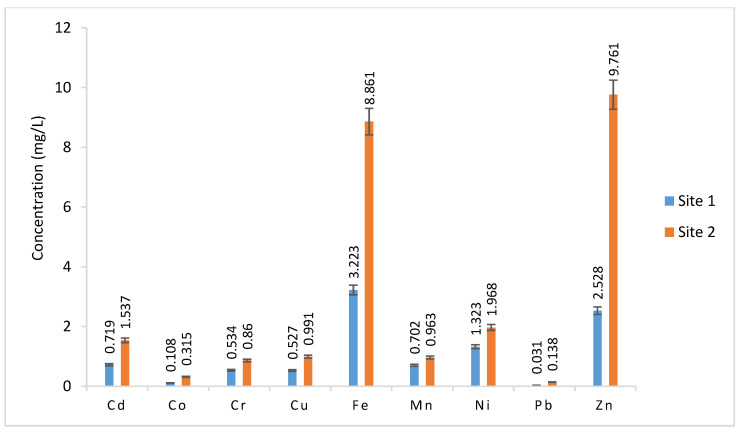
Metal values in water samples.

**Figure 3 plants-12-03652-f003:**
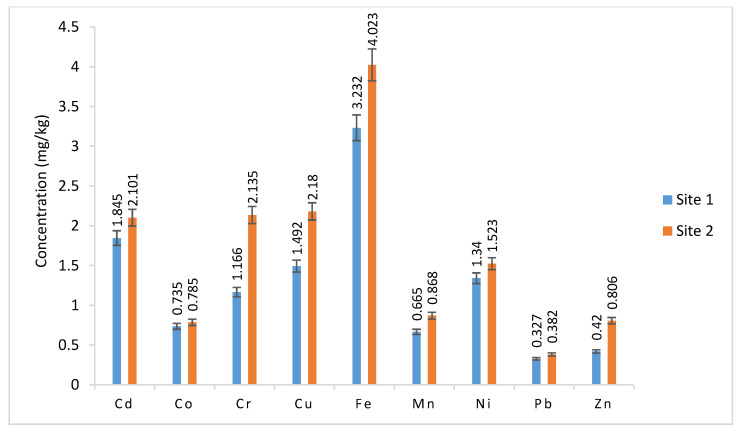
Metal values in soil samples.

**Figure 4 plants-12-03652-f004:**
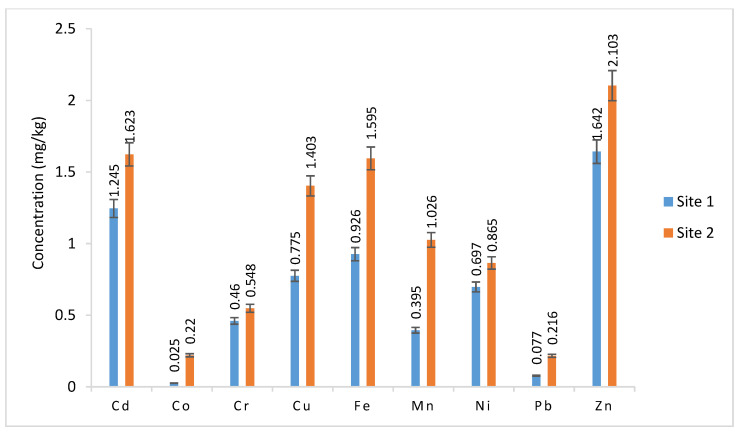
Metal values in coriander samples.

**Figure 5 plants-12-03652-f005:**
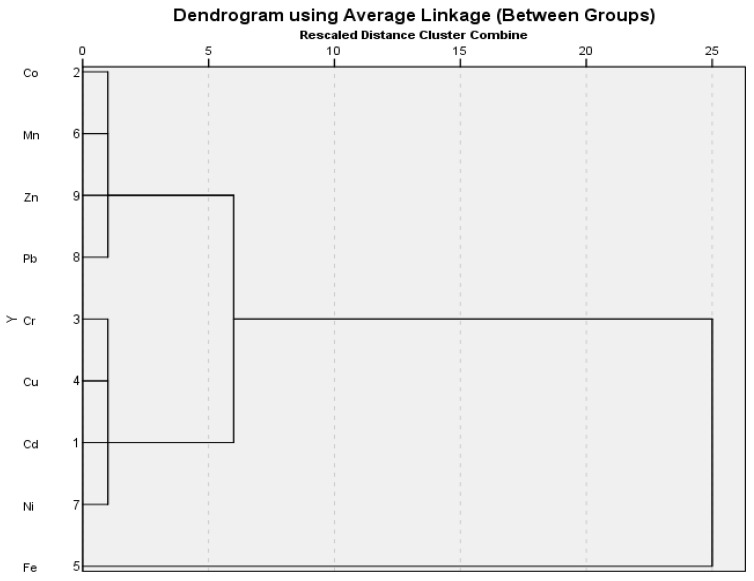
Dendrogram created using the soil sample values from two different locations.

**Figure 6 plants-12-03652-f006:**
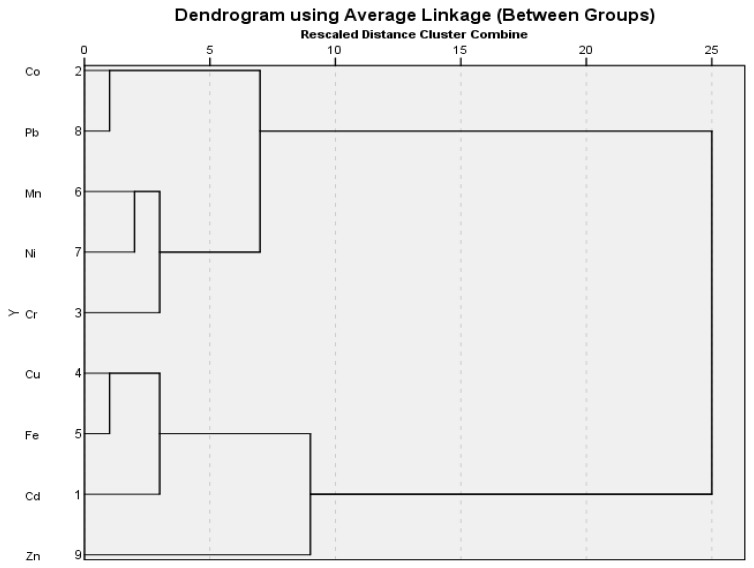
The dendrogram created using the coriander sample values from two different locations.

**Table 1 plants-12-03652-t001:** Operating conditions of flame atomic absorption spectrometry.

Element	Pb	Cd	Ni	Fe	Cu	Mn	Zn	Co
Wavelength (nm)	283.3	228.8	232.0	248.3	324.8	279.5	213.9	422.7
Slit width (nm)	0.7	0.7	0.2	0.2	0.7	0.2	0.7	0.7
Lamp current low (mA)	10	8	12	12	6	12	8	10
Acetylene flow rate (L/min)	2.0	1.8	1.6	2.2	1.8	2.2	2	2.8
Burner height (mm)	7	7	7	9	7	9	7	9

**Table 2 plants-12-03652-t002:** Analysis of variance for heavy metal values in water samples.

Source of Variation SOV	Degree of Freedom df	Mean Squares								
		Cd	Co	Cr	Cu	Fe	Mn	Ni	Pb	Zn
Treatments	1	0.355 ^ns^	0.003 ^ns^	0.003 ^ns^	0.597 ^ns^	8.394 *	0.216 ^ns^	0.841 ^ns^	0.003 ^ns^	0.007 ***
Error	6	0.045	0.095	0.002	0.065	0.634	0.045	0.506	0.018	0.002

* and *** significant at 0.05 and0.001; ns, non-significant.

**Table 3 plants-12-03652-t003:** Analysis of variance for metal values in soil samples.

Source of Variation	Degree of Freedom	Metal								
		Pb	Cd	Ni	Fe	Cu	Mn	Cr	Zn	Co
Sites	1	0.05 *	0.131 ^ns^	0.07 ^ns^	125.02 ^ns^	0.945 ^ns^	0.083 ^ns^	1.509 ^ns^	0.29 ^ns^	0.001 ***
Error	6	0.002	0.003	0.005	1.411	0.035	0.017	0.027	0.145	0.016

* and *** significant at 0.05, 0.01, and 0.001; ns, non-significant.

**Table 4 plants-12-03652-t004:** Physico-chemical parameters of the soil samples at two sites.

Physico-Chemical Parameters	pH	EC (dsm^−1^)	Organic Matter (%)	Texture Class
Site 1	7.8300 ± 0.0238	1.780 ± 0.023	0.550 ± 0.040	Loamy soil
Site 2	7.4500 ± 0.1322	5.090 ± 0.0267	0.690 ± 0.040	Loamy soil
MS	0.289 ^ns^	21.683 ^ns^	0.039 **	Loamy soil

** significant at 0.01; ns, non-significant.

**Table 5 plants-12-03652-t005:** Analysis of variance for metal values in coriander samples.

Source of Variation	Degree of Freedom	Metal								
		Pb	Cd	Ni	Fe	Cu	Mn	Cr	Zn	Co
Sites	1	0.03 **	0.27 ^ns^	0.05 *	0.89 ^ns^	0.79 ^ns^	0.79 ^ns^	0.01 **	0.42 ^ns^	0.07 ^ns^
Error	6	0.000	0.049	0.010	0.038	0.049	0.033	0.006	0.056	0.000

* and ** significant at 0.05 and 0.01; ns, non-significant.

**Table 6 plants-12-03652-t006:** Bioconcentration factor for metals.

Study Site	Metal								
	Pb	Cd	Ni	Fe	Mn	Cu	Cr	Zn	Co
Site 1	0.236	0.674	0.520	0.286	0.593	0.519	0.394	3.911	0.031
Site 2	0.565	0.774	0.567	0.396	1.181	0.643	0.269	2.609	0.280

**Table 7 plants-12-03652-t007:** Enrichment factor for metals in coriander.

Study Site	Metal								
	Pb	Cd	Ni	Fe	Mn	Cu	Cr	Zn	Co
Site 1	0.385	22.49	3.143	0.038	1.384	0.499	0.788	2.880	0.031
Site 2	0.621	25.75	3.428	0.053	1.717	0.991	0.539	1.921	0.080

**Table 8 plants-12-03652-t008:** Daily intake of metal and health risk index of coriander.

Study Site	Metal	Pb	Cd	Ni	Fe	Mn	Cu	Cr	Zn	Co
Site 1	DIM (mg/kg/day)	0.0003	0.132	0.0163	0.694	0.142	0.019	0.008	1.642	3.00
	HRI	0.078	132.8	0.813	0.992	3.465	0.494	0.005	5.475	0.001
Site 2	DIM (mg/kg/day)	0.0008	0.173	0.020	1.196	0.257	0.051	0.009	2.103	0.001
	HRI	0.220	173.2	1.009	1.709	6.276	1.282	0.006	7.012	0.006

## Data Availability

Not applicable.

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
