# Peer review of "Influence of Industrial Wastewater Irrigation on Heavy Metal Content in Coriander (Coriandrum sativum L.): Ecological and Health Risk Assessment"

_plants, 2023, doi:10.3390/plants12203652_

Round 1

Reviewer 1 Report

Dear Editor,

Thank you for your invitation to review plants-2639483 for Plants. Please find enclosed my comments.

Nowadays, unsafe food is a threat to consumers’ health and can effect economies globally. Effective food safety and quality control systems are safeguarding the human health and improving food markets. In general the manuscript is well written but the text of the manuscript needs corrections of English. The manuscript is sutible for Plants and I recommend accepting this manuscript after minor corrections. The minor corrections are explained in a following text, hoping the information provided by the reviewing will be helpful.

The submitted manuscript is technically correct and the research introduces investigation of the heavy metal contents in the water-soil- coriander samples in industrial wastewater irrigated areas, and to assess the health risks of these metal to humans.

First, it will be useful to point it out in Introduction not in Conclusion part that water from the sugar industry with heavy metals can influence on a metal content of plants like coriander, cauliflower, radish, lettuce, and cabbage. Since coriander is not eaten in the same amount as e.g. cabbage possible health risk is lower. Perhaps to mention that usually seeds are eaten (and leaves) as a spice? As the authors concluded, future research should be focused on the possible health risks associated with exposure to heavy metals through various pathways by better sampling plan.

rows 55-56: …and contain significant dietary minerals... Please explain what are dietary minerals - iron, magnesium, calcium and manganese are not?

Also sentences have to be rewritten: Coriander leaves are rich in vitamin C, vitamin A and vitamin K and contain significant dietary minerals.

However, seeds generally have higher vitamin content and contain significant amounts of iron, magnesium, calcium and manganese.

rich in vitamin C, vitamin A and vitamin K - have higher vitamin content – again C, A and K? or?

row 72: …450°F in the summer and 0°F in the winter… Why not in oC as in the rest of paper?

row 91: 25 samples of these vegetables were taken… the whole plant, leaves or just the seeds? At what time of the year or are they ready to harvest whole year?

row 95: Nitric acid (HNO3) was added… volume, concentrated or v/v?

row 97: Hydrogen peroxide (H2O2) was then… volume?

row 101: A sample of 1 gram of each vegetable – each vegetable, not only coriander?

row 118: S/n = 10 for LOD? LOQ?

row 161: RfD is ?

Table 2:  .355 please change in 0.355 (also for other values in table 2 and table 5)

row 199: As a result, this study supports earlier findings. – reference?

Minor editing of English language required

Author Response

added an attachment.

Reviewer 2 Report

Presented manuscript „Potential of industrial wastewater irrigation for heavy metal contamination in coriander (Coriandrum sativum L.): Ecological and health risk assessment“ has a potential  but in present form is my recommendation at least major revision.

Below is list of comments and remarks.

Abstract

Line 16 Why is concentration of metals in waters in mg/kg and not in mg/L?

2.1 Study area and Figure 1

In text authors mentioned districts (Jhelum, Mandi Bahauddin,….) and tehsils. Very useful will be if this information will be highlighted at the figure 1.

Also sampling area should be highlighted at the map.

Line 82 Site 1 and site 2 please show it on the map.

Line 82 „ Acid polypropylene  was used to wash the bottle,….“ Probably mistake. Really acid polypropylene?

Line 97 „…. And it was cooked in the digestive tube for an hour at 750 °C until the solution was clear.“  Really 750 °C? That was some high pressure, closed system for sample digestion?

Line 119 „ The presence of nickel and chromium was determined using very sensitive hydride technique.“ Really hydride technique (as gaseous nickel hydride and chromium hydride) or by technique of vapour compounds generation?

Line 129 „SRM NIST 1577b for vegetables“ this SRM is bovine liver – different matrix than plant material

And so and so..

It is not obvious from methodology if the waste water was analysed only once and the metal content is stable. Also, the description of the experiment needs to be improved. It is not obvious if one site was irrigated with ground water and second with waste water. There are many question for methodology. Was water for watering of coriander in some reservoir? Were the metals in the water measured only once? Or were metals monitored periodically?

Author Response

added an attachment.

Reviewer 3 Report

General comment: The authors investigated the heavy metal contents (Cd, Co, Cr, Cu, Fe, Mn, Ni, Pb, and Zn) in the water-soil-coriander samples in industrial wastewater irrigated areas and estimated the ecological and health risks due to heavy metals exposure. The authors claimed that the amounts of heavy metals in the soil and coriander samples irrigated with industrial wastewater were lower than the maximum permitted limits, except for Cd. The reviewer believed that the present study is interesting and potentially could contribute to the research field, however, there are some major concerns and questions that require be addressed to clarity and improve the present version.

Specific comments:

Title

The title seems confusing; therefore, it should be revised in a way that will give clear information and be relevant to the major findings.

Abstract

In the abstract, the aim of the study is clearly mentioned, but major results are not properly presented. Page 1, Lines 14-20, there is only one sentence and much information and data. So, it would be better to accommodate this information and data in several sentences. However, there is no information on the methodological parts of the study in the abstract. It should be included in the revised version of the manuscript.

Introduction

The introduction section does not provide sufficient background of the research topic. Please discuss the significance and novelty of the current investigation.

Materials and Methods

In general, not well explained. This section (materials and methods) has serious flaws. In figure 1 and in the text (line 78), sampling location coordinates should be included. In lines 81-82, the authors mentioned that they had tested groundwater from each source and wastewater from the sugar industry used to irrigate Sites 1 and 2, and their metal concentrations were determined. Thus, please discuss the heavy metal contents in groundwater in the results and discussion section. Otherwise, the authors should make it clear that Site 1 is irrigated using Groundwater and Site 2 is irrigated using wastewater. In line 85, the authors stated that “Twenty-five (25) soil samples were obtained at Sites 1 and 2”. Were 25 soil samples collected from each site or from both sites? If a total of 25 soil samples were collected from both sites, then how many from each site? Same questions for Coriander samples. Please discuss this clearly. In line 113, ‘2.5. Metal Analysis’ subsection, ‘heavy metal contents analysis in groundwater and wastewater’ must be included as the results showed in the results and discussion section.  

Results and discussion

The results section is also not well organized and not well explained. Please compare your results with other reports. In line 163, what does it mean by ‘3.1. Toxicity of heavy metals on water’. Please rewrite the subsection. Maybe ‘Heavy metal contents in irrigation water samples’ is better than ‘Toxicity of heavy metals on water’. Please rewrite/correct the same things in lines 182 and 221. Please discuss the probable sources of Cd in water, soil, and Coriander.  

The discussion could be more focused. The discussion doesn’t properly reflect the results obtained in this study. Authors can give more emphasis to discussing their findings from multiple angles.

Conclusion

The conclusion properly answers the aims of the study. Major limitations and opportunities to inform future research are not addressed properly. 

Overall comments: The manuscript is not well-written. Moderate English changes are required.

Moderate English changes are required.

Author Response

added an attachment.

Round 2

Reviewer 2 Report

Changes in the manuscript made the work clearer.

Reviewer 3 Report

All my previous remarks and comments have been considered in the new version of the manuscript. It means that the reviewed manuscript meets the minimum criteria, and in my opinion, can be published as an original paper in Plants Journal after a thorough author proofread.